

# Extracellular pH, osmolarity, temperature and humidity could discourage SARS-CoV-2 cell docking and propagation *via* intercellular signaling pathways

Franco Cicconetti[1], Piero Sestili[2], Valeria Madiai[3],
Maria Cristina Albertini[2], Luigi Campanella[4], Sofia Coppari[2],
Daniele Fraternale[2], Bryan Saunders[5,6] and Laura Teodori[3]

[1] Department of Emergency DEA-Surgery, University of Roma "La Sapienza", Rome, Italy
[2] Department of Biomolecular Sciences, University of Urbino, Urbino, Italy
[3] Laboratory of Diagnostics and Metrology, FSN-TECFIS-DIM, ENEA, Frascati-Rome, Italy
[4] Department of Chemistry, University of Roma "La Sapienza", Rome, Italy
[5] Applied Physiology and Nutrition Research Group, Universidade de São Paulo, São Paulo, Brazil
[6] Institute of Orthopaedics and Traumatology, Faculty of Medicine FMUSP, Universidade de São Paulo, São Paulo, Brazil

Corresponding author
Maria Cristina Albertini,
maria.albertini@uniurb.it

## ABSTRACT

The COVID-19 pandemic and its virus variants continue to pose a serious and long-lasting threat worldwide. To combat the pandemic, the world's largest COVID-19 vaccination campaign is currently ongoing. As of July 19th 2021, 26.2% of the world population has received at least one dose of a COVID-19 vaccine (1.04 billion), and one billion has been fully vaccinated, with very high vaccination rates in countries like Israel, Malta, and the UEA. Conversely, only 1% of people in low-income countries have received at least one dose with examples of vaccination frequency as low as 0.07% in the Democratic Republic of Congo. It is thus of paramount importance that more research on alternate methods to counter cell infection and propagation is undertaken that could be implemented in low-income countries. Moreover, an adjunctive therapeutic intervention would help to avoid disease exacerbation in high-rate vaccinated countries too. Based on experimental biochemical evidence on viral cell fusion and propagation, herein we identify (i) extracellular pH (epH), (ii) temperature, and (iii) humidity and osmolarity as critical factors. These factors are here in discussed along with their implications on mucus thick layer, proteases, abundance of sialic acid, vascular permeability and exudate/edema. Heated, humidified air containing sodium bicarbonate has long been used in the treatment of certain diseases, and here we argue that warm inhalation of sodium bicarbonate might successfully target these endpoints. Although we highlight the molecular/cellular basis and the signalling pathways to support this intervention, we underscore the need for clinical investigations to encourage further research and clinical trials. In addition, we think that such an approach is also important in light of the high mutation rate of this virus originating from a rapid increase.

# INTRODUCTION

The COVID-19 pandemic caused by the SARS-CoV-2 virus is currently afflicting the world population. The scenario epidemiologists are constructing is unfortunately forecasting a long stay (*Scudellari, 2020*). COVID-19 infection often leads to severe acute respiratory syndromes. However, extra pulmonary manifestations have also been described *e.g.*, hematologic, gastroenterological, renal, dermatologic, neurologic, and psychiatric complications (*AlSamman et al., 2020*). Indeed, it is now clear that this disease may turn into a dangerous systemic one involving several organs, which might become symptomatic even after the virus becomes inactive. To contain and combat the pandemic, the world's largest COVID-19 vaccination campaign is underway (*World Health Organization, 2020*), although the organizational machinery for vaccine supply, access and administration is not without issue, and several setbacks have occurred in several countries.

## Global outlook

As of July 19[th] there were 562,547 new daily cases and 10,779 new deaths worldwide. Furthermore 26.2% of the world population has received at least one dose of a COVID-19 vaccine (1.04 billion), whereas one billion have been fully vaccinated, with very high rates in countries like Israel, Malta, and the UEA. Conversely, only 1% of people in low-income countries have received at least one dose with examples of vaccination frequency as low as 0.07% in the Democratic Republic of Congo (*Worldometer, 2021*) Graphical representation of vaccination progress is accessible *e.g.*, at Our World In Data (*Ritchie et al., 2020*). It is thus of paramount importance that more research on alternate methods to counter the COVID-19 disease is undertaken and that can be implemented in low income countries. Furthermore, as breakthrough infections and infections from virus variants are occurring even in countries with good vaccination rates, the study of adjunctive therapeutic interventions is of paramount importance. It is also too early to know the duration of protection of COVID-19 vaccines, and more research need to be done on the vaccination efficacy on variants (*Harvey et al., 2021*). Due to the high rate of mutations and emerging new variants, vaccines may need to be adapted to ensure protection. Some data raise concerns on the level of protection provided by COVID-19 vaccines against some variants (*Focosi & Maggi, 2021*; *Krishnan & Krishnan, 2021*; *Chen et al., 2021*; *Baj et al., 2021*). Amino acid substitutions in the spike protein (S-protein) are an important virus strategy to evade the host immune response. Indeed, if the mutation is in the antigenicity region of the S protein, this may allow viruses to escape host antibodies and the genetic drift may represent a way for the virus to maintain its diffusion in the human population. Whether genetic variations affect the related antigenic phenotype will need to be confirmed by antigen analyses, using S genes with such mutants (*Ren et al., 2015*). Thus, SARS-CoV-2 entry inhibitors have an important role against

COVID-19 and research on therapeutic approached to that work alongside the vaccines are still needed.

In this scenario, the discovery of cheap and easily available medical devices that could help to prevent or ameliorate COVID-19 symptoms, thus avoiding advanced cases clogging the sanitary structures and intensive-care-units, would be a great resource. In addition, oxygen therapy intubation and mechanical ventilation is required when respiratory failure occurs, especially in compromised patients, and this may yield considerable health risks and higher frequency of death (*Sanchez et al., 2008*; *Khan et al., 2020*). The need for an accessible, economical and easy to handle approach is important as the pandemic is increasingly overwhelming the planet. Understanding virus behavior in relation to pH levels, temperature and other micro environmental factors is of paramount importance for virus inactivation and attenuation, vaccine formulation and quality control, and drug targeting (*Scheller et al., 2020*). Thus, SARS-CoV-2 entry inhibitors have an important role and COVID-19 and the research on therapeutics approach are still needed, to work alongside the vaccines and other drugs approaches (*Dong et al., 2021*).

## Background and scientific rationale

We have focused our review on the biochemical, physiological and physical cellular mechanism *via* which pH, osmolarity, humidity and temperature changes modulate the micro-environmental airway tract in limiting virus infection and proliferation, with the ultimate goal to discourage virus infection and modulate the cellular response against virus infection. One of the possible approaches to modulate airway environment is warm inhalation with sodium bicarbonate (SB). SB at 8.4% concentration is an alkaline solution of approximately 8.5 pH, and is an alkalinizing agent widely used in the treatment of metabolic acidosis (*Drugs.com, 2021*; *Integrated Biomedical Technology, 2003*) (see below). Although no consensus currently exists on the utility of alkaline therapy for respiratory acidosis, very little attention remains dedicated to this approach. Undoubtedly, controlled studies are necessary to test the efficacy and efficiency of alkaline therapy to establish the optimal mode of administration (dose, rate, and tonicity). However, most of the controversies on alkaline therapy are based on intravenous administration of SB (*Chand, Swenson & Goldfarb, 2021*).

Based on experimental data, we have outlined the cellular/molecular mechanisms that work against the virus in the presence of SB warm vapour. This intervention could be a highly valuable, safe, well-tolerated and cost-effective treatment as reported below, although clinical information is scarce regarding the potential application of such an approach, thus researchers may be unaware of its potential as an adjunct preventive treatment in the fight against COVID-19. Therefore, herein we provide appropriate information from the literature to provide a theoretical basis to propose SB warm inhalation therapy as a worthwhile tool for further investigation in the fight against COVID-19. To the best of our knowledge, few clinical trial has been performed (*US National Library of Medicine, 2020*; *Mody, 2020*), although trials exist for other pathologies. Results of SB clinical trials (although not *via* inhalation) for

acidosis-associated diseases, such as diabetes, severe dehydration and renal insufficiency, have been reported (*Melamed et al., 2020*; *Quade, Parker & Occhipinti, 2021*). At present, SB inhalation can be used as a therapeutic agent in the treatment of acidosis caused by cystic fibrosis (CF) (*Gomez et al., 2020*). The study reports that inhalation of aerosolized SB on 12 CF volunteers is secure (no side effects were evidenced) and well tolerated even at the highest dose (five mL saturated solution, pH 8.4%, twice a day). The benefits of inhaled $NaHCO_3$ are due to the increase of pH of the airway surface liquid (sputum rheology improvement, sputum viscosity and viscoelasticity reduction). These changes may be responsible of enhancing the respiratory immune defense favoring mucociliary clearance to maintain airway hygiene. Other studies have demonstrated that the use of hypertonic saline solution is an inexpensive, safe, and effective additional therapy in CF patients (*Taylor & Kuhn, 2007*). Recently, an *in vitro* study provided molecular evidence that SB may have a direct therapeutic effect on the bronchial epithelium (*Gróf et al., 2020*) while confirming the safe therapeutic use of inhaled sodium bicarbonate. A clinical study on COVID-19 (*Wardeh, Conklin & Ko, 2020*) reported a significant improvement of COVID-19 patients after SB inhalation. Improved clinical parameters were observed also in another clinical study (*Mody, 2020*).

As with other viral infections, SARS-CoV-2 and their variants thrive in specific environmental conditions that optimize virus fusion. A mechanistic explanation of why the therapeutic approach of SB warm inhalation may provide a hostile and detrimental environment for the virus is discussed herein. SB-modulated environmental factors which affect viral fusion include (i) external pH (epH), (ii) temperature, (iii) humidity and osmolarity. These factors are discussed along with their implications on mucus thick layer, proteases, abundance of sialic acid, vascular permeability and exudate/edema.

## SURVEY METHODOLOGY

Papers found in Pubmed describing cellular/molecular mechanisms to support the use of warm SB inhalation were considered. In particular, whenever available, data focused on chemical/physical features such as pH, osmolarity, and humidity involved in SARS-CoV2 docking, replication and propagation steps were included, together with the activated cell signaling. In case no published data on SARS-Cov2 were available, results regarding other coronaviruses or similar viruses were discussed. Additionally, due to the overwhelming publication rate, we also consulted the new COVID-19 platform resource, CoronaCentral Dashboard (https://coronacentral.ai/) (*Lever & Altman, 2021*).

Papers with controversial, conflicting and opposing opinions/data were also considered to provide a comprehensive and balanced overview.

### Increased external pH may disadvantage virus infection

Extracellular pH (epH) affects many cellular/molecular structures including lipid bilayers (*Yamaguchi et al., 1982*; *Angelova et al., 2018*), protein expression (*Olson, 1993*) and even intracellular pH (*Fellenz & Gerweck, 1988*). pH is a critical factor whose change gives rise to significant alterations in the protein structure. Conformational change of viral proteins involved in docking and replication in the host cell could yield virus

inactivation. Indeed, epH is one of the most important environmental conditions influencing a virus's infection (*Grinstein, Swallow & Rotstein, 1991*; *Slonczewski et al., 2009*; *Abou Alaiwa et al., 2014*; *El Badrawy et al., 2018*; *Stancioiu et al., 2020*). Cell membranes fusion of to the influenza virus is promoted by an appropriate acidic epH depending on the virus (*Helenius, 2013*).

SARS-CoV-2 shares many of the characteristics of coronavirus and the general mechanism for SARS-CoV-two infection has been identified on the basis of knowledge of the other SARS-CoV and MERS-CoV. In particular, coronaviruses are stable at pH 6.0 and 37 °C (half-life 24 h), but are quickly and permanently inactivated by short-term treatment at pH 8.0 and 37 °C (half-life 30 min) (*Sturman, Ricard & Holmes, 1990*).

Low pH is necessary for conformational changes (activation) of viral-glycoproteins (*Dollery, Delboy & Nicola, 2010*; *Yuan et al., 2018*). The S glycoprotein protein is the "viral armed wing" of a glycoprotein. Proteolytic activation of viral glycoproteins by endosomal proteases also needs low pH (*Yuan et al., 2018*). It is well known that the first stage of infection for Coronaviridae is S-protein-mediated attachment to the cell surface receptor. However, SARS and other human coronaviruses may use redundant mechanisms for cell docking *e.g.*, ACE2, N-acetyl-9-O-acetyl neuraminic acid (sialic acid: SA) moieties and heparan sulphate, thus many recent studies focused on molecular modelling of SARS-CoV2-host docking. Many of these docking mechanisms are pH dependent. So far, studies show that SARS-CoV-2 uses the ACE2 receptor as a main docking protein (*Yang et al., 2020*; *Teodori et al., 2020*). ACE2 is essential for SARS-CoV-2 fusion, but it is uncertain whether ACE2 interactions are enough for SARS-CoV-2 binding. ACE2 is probably necessary for the entry of the virus but might not be the unique or primary cell surface binding site. Heparan sulfate proteoglycans act as adhesion molecules, perhaps making the interaction between SARS-CoV-2 and ACE2 easier. Thus, infection modalities are more complicated than so far reported and a more appropriate knowledge of the infection process is necessary for new drug discoveries (*Gallagher et al., 2013*; *Zamorano Cuervo & Grandvaux, 2020*).

Sialic acids (SAs) are an important class of receptors for several human viruses infection, including SARS-CoV binding to respiratory tract epithelium cells (*Bouvier & Palese, 2008*). SA-receptors are also important for CoVs docking, and the binding of the virus to the host cell are pH-mediated (pH 5–6). Recently, it has been demonstrated that the S-protein from SARS-COV-2 binds SAs (*Baker et al., 2020*; *Milanetti et al., 2020*). Adequate virus-membrane fusion (*i.e.*, fusion pore efficient forming and delivery/transport of the viral RNPs into the nucleus) is necessary for new viral RNA replication, transcription and translation of the viral proteins and formation of new viruses.

For several viruses, acidification of the capsid is critical for viral entry. In these cases, viral capsids show protease active sites sensitive to pH. In addition, acidic pH similar to endosomes' pH induces a structural change in the capsid that induces autolytic protease activity and this pH-dependent protease activity may be important for viral infection (*Salganik et al., 2012*). The pH-activated proton-selective channel M2 also has an important role in virus replication (*Takeda et al., 2002*). The M2 proton channel responds to epH; specifically, low epH activates the channel and high epH closes the channel. M2

mediates membrane scission during the budding of influenza viruses (*Holsinger et al., 1994*). During infection, increased glycolytic activity of infected cells produces an increased release of $H^+$ from the infected cells endosomes through the viral M2 $H^+$ channel, leading to a lowered epH at the cell-surface (*Liu et al., 2016*). These mechanisms support viral reproduction. To our knowledge, no investigations exist on ion-transmembrane-exchange-proteins modulation in SARS-CoV-2, although this is an interesting topic to explore. Indeed, diseases of ion channel function, such as cystic fibrosis, lead to dysregulated fluid levels in various lung compartments, and these diseases are often associated with pulmonary infection (*Rowe, Miller & Sorscher, 2005*). Protease machinery is also highly epH-dependent, as in the case of TMPRSS2 (transmembrane-protease-serine-2) for S-protein priming. The TMPRSS2 is a proteolytic enzyme that forms part of the ACE2 receptor and has been identified as fundamental. TMPRSS2 acts on protein S at the S1/S2 cleavage site, detaching the S1 subunit and thus ensuring fusion with the cell membrane. The proteolytic cut causes a conformational change that causes the protein to open, resulting in the fusion peptide approaching the cell membrane. Subsequently, the S2 subunit of the S protein, closes on itself, causing the formation of a pore which leads to the completion of fusion (*Nieto-Torres et al., 2014*).

TMPRSS2 virus entry can be blocked by inhibitors of cellular TMPRSS2 (*Yamaya et al., 2016*). TMPRSS2 might also promote viral diffusion and pathogenesis by diminishing viral neutralizing antibodies recognition and activating CoV S-protein for virus-cell fusion (*Glowacka et al., 2011*). TMPRSS2 is present in airway epithelial cells; it triggers the fusion of the viral and endosomal membranes and has an optimal activity at acidic pH. Priming of S-proteins by target cell proteases is fundamental for viral entry and cleavage at the S1/S2 and the S2' sites. The S1/S2 cleavage site of S-protein fosters many arginine residues, indicating a high cleavability. Noteworthy, the cleavage site sequence can determine the coronaviruses zoonotic potential. Indeed, host TMPRSS2 involvement in viral diffusion has been described for COVID-19 (*Mollica, Rizzo & Massari, 2020*; *Zipeto et al., 2020*).

In addition to the influence of epH on SAs-receptor docking, it can modify SA activity. Virus neuraminidase (NA), a sialidase that is one of the major surface glycoproteins of influenza viruses, needs an acidic environment to degrade SAs. The importance of SAs on SARS-CoV-2 has been shown (*Kim, 2020*). Due to the ubiquitous distribution and location of SAs, they modulate many cellular functions and pathological processes. SAs are also the binding targets of a great number of pathogenic organisms and their toxins. NA and hemagglutinin are two relevant virus antigens. The main functions of NA are the cleavage of SAs from the host cells surface, aiding in the release of the new virus produced by the infected cells and aiding virus transport through the sialated mucus, present in the airway tract. Although there is scant information on the binding affinity of SAs and NA, SA-binding and NA-cleavage are important factors for infection progression. Both acidic epH and physiological temperature are necessary for good NA enzymatic activity (*Garcia et al., 2014*).

Other studies suggest a host-variable protective role of SAs in secreted mucus (*Zanin et al., 2016*). Respiratory mucus traps and neutralizes viruses (*Eccles, 2020*). Pathogen

binding can be blocked by soluble mucins, secreted into airways which contain large quantities of SAs. This complexity needs to be considered to understand the CoV pathogenesis processes.

Little attention has been paid to the influence of epH on the immune response (*Kellum, Song & Li, 2004*). Environmental pH influences multiple immunological functions, unravelling an interplay between epH and immune cells (*Díaz, Dantas & Geffner, 2018*). Extracellular acidosis is a hallmark of the inflammation processes. Accumulation of protons in the extracellular environment is associated with the inflammatory course. High concentrations of protons are recognized by innate immune cells as a "danger-associated molecular pattern" (*Casimir et al., 2018*). Recent evidence suggests links between acid-base balance and cytokine concentrations, with a certain level of acidosis triggering the inflammatory condition. When acute infection is activated, the phagocyte-based innate immune system plays a fundamental role since, the proton concentration may provoke inflammation (*Casimir et al., 2018*). Proton-sensing G protein-coupled receptors (*e.g.*, GPR4) are important for pH homeostasis and inflammation control (*Kellum, 2002*). Higher expression of GPR4 is observed in patients with inflammatory disease (*e.g.*, bowel inflammation; *de Vallière et al., 2015*). GPR4 is upregulated by an acid epH and downregulated at pH 7.4, with very little activity observed at a more alkaline pH (*e.g.*, >7.4) (*Chen et al., 2011*; *Dong et al., 2013*). Increased expression of many inflammatory genes (*e.g.*, chemokines, cytokines, adhesion molecules, nuclear factor kappa B pathway genes, prostaglandin-endoperoxide synthase 2, and stress response genes) is observed when GPR4 is over activated. Acidosis, together with other environmental factors, may act as a regulator of the immune response able to induce a pro-inflammatory or a pro-resolving immune response depending on the context (*Díaz, Dantas & Geffner, 2018*).

Many stages in the viral replication life cycle are dependent on a low epH, and the SARS-CoV-2 virus appears no different. Thus, to combat viral progression, inhalation of aerosolized SB could be an interesting preventive tool since it will increase the epH creating an inhospitable environment for virus docking and replication. There are several experimental studies on the influence of epH and modification of cell signalling and cell physiology that are connected with virus infection, thus supporting our hypothesis. Studies on SARS-COV-2 infection and epH are currently ongoing and slowly being published. A recent paper with the same hypothesis has now been published (*Wang, 2021*), further strengthening our proposal. Indeed, the authors hypotize that enriching the environment of several negatively charged ions, including *e.g.*, $O_2^-$, $O^-$, $O_3^-$, $CO_3^-$, $HCO_3^-$, $NO_3^-$, $NO_2^-$, and $OH^-$, thus increasing epH, would discourage virus fusion. Nonetheless, we believe the benefits of warm SB inhalation is not exclusively on epH changes, but also other physicochemical characteristics such as humidity and temperature, as reported below.

### Increased extracellular temperature may disadvantage virus infection

Together with pH, temperature is another innate physiological barrier against infections. Many microorganisms do not survive beyond certain temperatures. In addition,

temperature exerts substantial mechanical and biochemical effects on the airway epithelium.

The lung tissue represents the largest surface area of the human body exposed to the external environment (*Karamaoun et al., 2018*). A large area of the bronchial epithelium is sheltered by a mucus layer to protect against foreign particles and pathogens. This mucus layer and periciliary layer are thick gel-like substances made of water and mucins. The cilia present in the epithelium, by beating metachronously in the pericilia layer, can transfer the mucus to the top of the trachea to be cleared. This mucociliary transport and clearance is a fundamental defense mechanism of the lungs against invading pathogens and is a major first line of defense against respiratory pathogens. Many pulmonary diseases are associated with an impairment of the mucociliary clearance and accumulation of mucus in airway lumen. Although several aspects of the bronchial mucus dynamics still remain unclear, it has been shown that the temperature and relative humidity (discussed below) of the inspired air can influence the efficiency of mucus clearance (*Saketkhoo, Januszkiewicz & Sackner, 1978*; *Boucher et al., 1981*; *Williams et al., 1996*; *Karamaoun et al., 2018*). Temperature in the respiratory tract is an important environmental factor influencing virus infection and host reaction. For example, some common cold viruses (*e.g.*, rhinovirus) replicate more efficiently at 33–35 °C than at 37 °C temperatures of the nasal cavity environment. A less robust interferon (IFN) and IFN-stimulated-gene response is observed in respiratory epithelial cells at cool temperatures (*Foxman et al., 2015*). Indeed, higher temperature induces higher expression of type I and type III IFN genes and IFN-stimulated-genes in host cells of the respiratory tract resulting in a generalized antiviral resistance. CoV induces a higher IFN-dependent innate immune response at 37 °C than at 33 °C. Experimental data demonstrate that SARS-CoV-2 also induces type I IFN expression following infection of macrophages. Other investigations have shown that induction of temperature-dependent IFN following rhinovirus infection depend on the mitochondrial antiviral-signaling (MAVS) protein, a key signalling adaptor of the RIG-I–like receptors. RIG-I is a sensing molecule able to distinguish viral dsRNA from host dsRNA. Once the sensor is activated, it interacts with MAVS (mitochondrial antiviral) which, once activated, continues the signalling cascade through the kinases TBK1 and IKKε. In addition, IKKε signalling also induces many IFN-inducible proteins *via* the STAT1 pathway. The genes induced in response to IFN-β, trigger the cell to block virus replication and induce antiviral protein secretion in response to secreted IFN (*Frieman & Baric, 2008*). Some studies have also demonstrated that SARS-coronavirus suppresses innate immunity by mitochondria and targets MAVS/TRAF3/TRAF6 signalling. This severely limits host cell IFN responses (*Shi et al., 2014*). IFN exerts its antiviral effects by inducing more than 300 IFN-stimulated genes with diverse antiviral functions. SARS-CoV-2 infected cells have an impaired interferon response suggesting that the virus disrupts the normal host cell interferon response (*Schneider, Chevillotte & Rice, 2014*).

Advanced-stage COVID-19 patients with respiratory symptoms undergo oxygen ventilation (*Grasselli et al., 2020*). This exposes patients to the inhalation of cold air which produce pro-inflammatory substances (*D'Amato et al., 2018*). Exposure to cold air

increases macrophage and granulocyte numbers in the lower airway tract (*Larsson et al., 1998*). Moreover, cold-related dysregulation of respiratory mucociliary function inhibits microorganisms and pollutants clearance (*Clary-Meinesz et al., 1992*). Thus, in addition to its effects on epH, warm SB inhalation might provide a sub-optimal environmental temperature for virus replication and serve to improve some of the respiratory mucociliary function impairment associated with cold air inhalation. Intubation/ventilation may be overused for COVID-19 patients and simpler and more widely available devices are desirable which can similarly treat early-stage patients to avoid disease progression and mitigate symptoms.

Throughout the preparation of this review paper, trials using steam inhalation for thermal inactivation of SARS COVID-2 virus are being conducted and providing encouraging results (*la Marca et al., 2021*).

## Increased external humidity and osmolarity may disadvantage virus infection

Virus survival and transmission also depend on humidity. Indeed, humidity and water exchange between mucus and inhaled/exhaled air (evaporation/condensation) also influence mucociliary clearance alone or in conjunction with temperature (*Bustamante-Marin & Ostrowski, 2017*). It is well-known that as the air became drier, mucociliary clearance slows. The influence of humidity levels on the mucous membrane environment in animal models has been investigated *in vitro* at several temperatures, showing that the mucociliary wave frequency was decreased in reduced environment humidity from 90% to 20% (*Mercke, 1975*).

Pathologic processes of the upper-respiratory-tract (mucosal congestion and edema) could be influenced by mucosal microcirculation modulation controlling hypertonicity. Reasonable evidence shows that the airway epithelium fluid layer is exposed to changes in tonicity. Abnormalities in the homeostasis of the airway surface liquid layer lead to consequent failure in maintaining an adequate lung defense. Abnormal airway surface fluid respiratory tract epithelia fail to kill bacteria. Hypertonic saline affects mucociliary clearance and clinical outcomes in chronic bronchitis (*Wark & Mcdonald, 2018*; *Bennett et al., 2020*). Together with the biochemical effects described above, SB warm inhalation may also provide a physical effect as an osmotic pump. SB warm inhalation may create favorable osmotic conditions to repair damages from oxygen demand. The inhalation of steam may alleviate the constriction of lung and air sacs, allowing mucus and organisms to move out of the lungs more readily, and allow oxygen to diffuse into the lungs more efficiently. In a review of 19 trials, that hypertonic saline solution (HS) (nebulised as a fine mist through a mask or mouthpiece) appeared to be an effective adjunct therapy during acute exacerbations of lung disease in adults (*Wark & Mcdonald, 2018*). In another study, inhaled HS was delivered to people with CF to promote mucus clearance *via* an increased ionic strength (*Alaiwa et al., 2016*). Ongoing research using steam inhalation therapy with the addition of natural products against COVID-19 are also

**Proposal of intervention study**

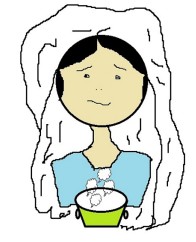

| Extracellular PH modification | | Osmolarity hypertonic extracellular enviroment |
|---|---|---|

**WARM SB INHALATION**

| Extracellular temperature increase |
|---|

**Figure 1 Warm sodium bicarbonate (SB) inhalation modifies airway microenvironment fighting COVID-19 progression.** The treatment contemplates, twice daily, one L of water with 20 g of SB that should be brought to the boil. When heat has been turned off, the head should be placed at 30–40 cm covered with a cloth to breath normally for 15 min.

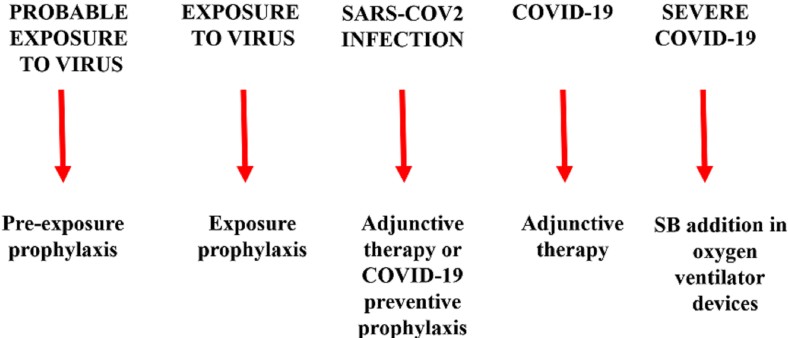

**Figure 2 Warm sodium bicarbonate (SB) inhalation therapy steps that may help to contrast SARS-COV-2 infection and COVID-19 progression.** The therapy may be used as pre-exposure/exposure prophylaxis, as adjunctive therapy (either during SARS-COV2 infection and COVID-19 disease progression) or added in oxygen ventilator devices for severe COVID-19 patients.

being published (*Chakraborty & Daniel, 2020*; *la Marca et al., 2021*; *Shanmugaraj et al., 2021*).

Warm SB steam inhalation therapy may be considered a safe medical device against COVID-19 progression provided that some conditions are followed to avoid irritation or scalding injuries, although these are very unlikely (*Eschenbacher et al., 1991*). Approximately one L of water should be brought to the boil before adding 20 g of SB. After turning off the heat, the head of the patient should be placed 30–40 cm above the solution and covered with a cloth. The individual should then breath normally for 15 min (Fig. 1). This therapy may be used as pre-exposure/exposure prophylaxis, as adjunctive therapy (either during SARS-COV2 infection and COVID-19 disease progression) or added in oxygen ventilator devices for severe COVID-19 patients. This protocol should be implemented twice daily (Fig. 2).

## CONCLUSION

Herein we have illustrated a rationale for combatting COVID-19 by modulating the physicochemical microenvironment of the airway tissue which represents the first route of virus infection. One method of achieving such a micro-environmental modulation is through the administration of nebulized warm SB inhalation.

This proposal is in line with the "COVID Action Platform" of the World Economic Forum to perform new evidence-based approaches: "*to potentially mitigate the risk*" (*World Economic Forum, 2020*). Indeed, this approach might represent a pleiotropic adjunctive strategy to limit SARS-CoV-2 infection/replication and disease progression within host airways. Based on the molecular/cellular evidence herein above, warm SB inhalation could be provided as a valuable preventive prophylaxis alongside other preventive approaches. Such an approach could also be administered in the early stages of the malady, to avoid the escalation of symptoms and the development of the most severe, life-threatening stages. Furthermore, warm SB inhalation could also be provided alongside oxygen therapy, appropriately modifying the ventilation devices

Most importantly, we highlight the possibility that SB warm inhalation could discourage virus host cell adhesion and proliferation and avoid that the harshest outcome of the disease occurs. Indeed, SB warm inhalation creates a hostile physicochemical environment to undermine virus stability.

Warm SB inhalation currently remains a poorly explored, yet potentially underrated and underappreciated option that warrants further study, but and it could yet play an important role in the fight against COVID-19 disease progression. Indeed, further studies are needed to prove thatmodulating these cellular biochemical/physical features could be efficacious and safe against COVID-19 *in vivo*. Studies with CF suggest this therapy to be safe, although too few studies are available to quantify the effectiveness of this intervention, but the few reported above are encouraging. Early and timely treatment with therapeutic regimens controlling virus replication and inflammation might help to modify the course of disease progress, improve patients' recovery rate and time, ultimately avoiding the risk of hospital collapse sadly experienced even by most western countries. SB inhalation can easily be performed in a domiciliary care setting and could reasonably fulfil the extraordinarily urgent need for early, simple and cost-friendly interventions. Simple SB inhalation may be effective to prevent or mitigate COVID-19 symptoms. This virtually cost-free modality would appreciably benefit the general population, not only for pre-exposure prophylaxis, but also in protecting against SARS-CoV-2 antigenic drift and future virus pandemics.

## ACKNOWLEDGEMENTS

The authors would like to warmly thank Prof. Bruno Gualano from the Applied Physiology & Nutrition Research Group, School of Medicine, University of Sao Paulo (Brazil), for his generous contribution to the manuscript.

The authors would also like to thank Fausto Tili (Terme Francescane srl, Spello-PG, Italy) for his advice on bicarbonate water inhalation therapies.

### Funding

Bryan Saunders was financially supported by Fundação de Amparo à Pesquisa do Estado de São Paulo (2016/50438-0) and a grant from Faculdade de Medicina da Universidade de São Paulo (2020.1.362.5.2) The funders had no role in study design, data collection and analysis, decision to publish, or preparation of the manuscript.

### Grant Disclosures

The following grant information was disclosed by the authors:
Fundação de Amparo à Pesquisa do Estado de São Paulo: 2016/50438-0.
Faculdade de Medicina da Universidade de São Paulo: 2020.1.362.5.2.

### Competing Interests

Maria Cristina Albertini is an Academic Editor for PeerJ.

### Author Contributions

- Franco Cicconetti conceived and designed the experiments, performed the experiments, analyzed the data, authored or reviewed drafts of the paper, and approved the final draft.
- Piero Sestili analyzed the data, authored or reviewed drafts of the paper, and approved the final draft.
- Valeria Madiai performed the experiments, analyzed the data, authored or reviewed drafts of the paper, and approved the final draft.
- Maria Cristina Albertini analyzed the data, prepared figures and/or tables, authored or reviewed drafts of the paper, and approved the final draft.
- Luigi Campanella analyzed the data, authored or reviewed drafts of the paper, and approved the final draft.
- Sofia Coppari analyzed the data, prepared figures and/or tables, authored or reviewed drafts of the paper, and approved the final draft.
- Daniele Fraternale analyzed the data, authored or reviewed drafts of the paper, and approved the final draft.
- Bryan Saunders conceived and designed the experiments, performed the experiments, analyzed the data, authored or reviewed drafts of the paper, and approved the final draft.
- Laura Teodori conceived and designed the experiments, performed the experiments, analyzed the data, prepared figures and/or tables, authored or reviewed drafts of the paper, and approved the final draft.

### Data Availability

This is a literature review with no raw data.

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
