# Peer review of "Extracellular pH, osmolarity, temperature and humidity could discourage SARS-CoV-2 cell docking and propagation via intercellular signaling pathways"

_PeerJ, doi:10.7717/peerj.12227_

## Round 0.1 · original submission · Major Revisions

Please carefully address critiques of all reviewers and revise your manuscript accordingly.

Reviewer 1 ·

Basic reporting

The literature review article titled Extracellular pH, osmolarity, temperature and humidity can discourage SARS-CoV-2 cell docking and propagation via intercellular signaling pathways (#61344) by Cicconetti et al., describes 3-4 factors such as extracellular pH, temperature, humidity and osmolarity in different circumstances and drawn parallels to the SARS-CoV-2 disease progression and control.

Overall, the review article is well written on the chosen topics and the authors has made a sincere effort in finding the relevant literature. The review has scope and relevance in the current pandemic scenario and would benefit clinicians and scientists to explore other means of interventions. The review article is also with in the scope of the journal and this topic has not been reviewed as far as my knowledge is concerned. However, I have the below comments and concerns regarding the article.

Experimental design

The study design i.e., survey methodology is clearly demonstrating the outline of the review.

There are certain areas where more references are required which has been mentioned in the comments.

Overall the review is well organized. A few suggestions of various other related topics such as other Alkalinizing agents, other ssRNA virus and model figure showcasing the various factors regulated by sodium bicarbonate were made in the comments.

As a reviewer from a reader point of view the below comments were made to improve this review article.

Validity of the findings

Few conclusions were not supported by experimental works which need to be addressed by the authors. The future directions and conclusions are satisfactory in general.

Additional comments

Minor and Major changes suggested:
1. Line 32: Please change Countries to countries.
2. Line 33-34: Why only few countries are mentioned in the 4.5% of world population that has received vaccination? Rationale for the same.
3. A table representing the % vaccinated in various countries would be useful for the reader to understand how well the vaccination campaign has reached geographically
4. Line 33 and 67: Change 4,5% to 4.5%
5. Line 47-49: “A cheap…” sentence is not in line with the text flow. Please move it out from abstract so that the text flow is in line with the previous sentences
6. Line 45 and throughout the manuscript text: Please change “Signalling” to “Signaling”
7. Line 50: Please modify to “Virus originating from a rapid increase”
8. Line 69: The authors have mentioned “Like all vaccines”- please mention a few which are 100% effective and provide citations
9. Line70: How can the authors state that “covid-19 vaccines will not be 100% effective”. Please modify and provide references of the same.
10. Line 75: The authors has cited an unpublished article that is currently in MedRxiv. How can this conclusion be verified? Please provide a valid citation or mentioned unpublished results.
11. Line 77: “Initial data”: What is this data? please cite a reference to which study you have referred here.
12. Line 79: S protein: Elaborate on the first appearance of the text. It is written in line 80.
13. Line 89: Please change to “The need for an approach” and “economical”
14. Line 90: Please change to “Behavior”
15. Line 100: The authors has stated “Based on clear experimental published data” where is it published? No citation? And no data reference in the text. Please add
16. Line 114-116: Discuss the clinical study data briefly on the outcomes (statistics, % improvement etc)
17. Lines 142-143: References? “By an appropriate acidic pH” what is this pH value?
18. Line 148-149: References?
19. Line 150: Change to “needs low pH”
20. Line 156: Reference for CoV-2 uses ACE2 receptor
21. Line 164: Change to “Human virus infections, including”
22. Line 166: Change to “host cell;”
23. Line 219: Change to “quantities”
24. Line 229 and 231: References?
25. Line 234-237: Reference?
26. It would be interesting to the readers if the authors include details regarding other alkalinizing agents that are currently in medical use and their implication on SARS-CoV-2? A table representing the same would be helpful or few sentences. Also, what are the advantages of SB over other agents.
27. As the authors has mentioned few factors like temperature, epH etc that influence the CoV-2 infections, a model figure illustrating the same and their regulation would be helpful to easily understand the benefits.
28. Line 254, 255: Please change to “Defense”
29. The SARS-CoV-2 is a positive sense-ssRNA virus. It would be nice if the authors could look in literature about the affect of these factors (pH, temp..) on other Positive sense-ssRNA viruses and draw parallels to the SARS-CoV-2
30. Line 271: Elaborate “MAVS protein” in its first appearance in the text
31. Line 278: Change to “targets”
32. Line 288-291: The authors has drawn conclusions for the warm SB applicability but there are no experimental study data of its application in SARS-CoV-2. They have discussed SB role in various other circumstances but not directly to SARS-CoV-2. Please modify the sentence so that it remains as a hypothesis which needs to be investigated in clinical trials.
33. Line 288-291: Also, it would be useful if the authors can shed some light on how SB can be administrated, dosage, how to warm it and the possible side effects (any references of the same)
34. Line 299: Please change to “becomes”
35. Line 307: Please change to “Defense”
36. Line 311: Please change to “favorable”
37. Line 329-331: Please provide an exact link to the covid action platform-where the argument is in line with the proposal
38. Line 334: The authors has not mentioned that warm SB inhalation can be a preventive measure. Rather they have discussed it in terms of control of the virus propagation after infection. Please modify the sentence
39. Line 342: There is no clinical trial or experimental data is shown/cited regarding direct application of SB in Sars-CoV-2 but then how can the authors state that it is an underrated and underappreciated option, please modify the sentence.

Reviewer 2 ·

Basic reporting

Clear and unambiguous, professional English used throughout.

No, the article is not written in professional English. Some examples: line 86; lines 128-135 etc.

Is the review of broad and cross-disciplinary interest and within the scope of the journal?

Yes, the review is of broad and cross-disciplinary interest as focuses on different biochemical properties that may be important for SARS-CoV-2 and most other human viruses for cell entry.

Has the field been reviewed recently? If so, is there a good reason for this review (different point of view, accessible to a different audience, etc.)?

Yes, a recent publication reviewed recently a possibility of adjusting extracellular pH to prevent entry of SARS-CoV-2 into human cells.

Bin Wang. Adjusting extracellular pH to prevent entry of SARS-CoV-2 into human cells. Genome. 64(6): 595-598. https://doi.org/10.1139/gen-2020-0167

Current review by Cicconetti et al. doesn’t provide a different point of view. Also, the review fails to provide sufficient evidence on benefit and safety of the proposed methods, as such methods have not been shown to be effective against any other viral infection that depend on pH, temperature and osmolarity for infection.


Does the Introduction adequately introduce the subject and make it clear who the audience is/what the motivation is?

No, the authors do not provide sufficient background on the field including available vaccines and therapies available against SARS-CoV-2.

Experimental design

Is the Survey Methodology consistent with a comprehensive, unbiased coverage of the subject? If not, what is missing?

Survey Methodology needs to be revised as the language is not scientifically sound. Also, not sufficient literature review has been done in the field to propose inhalation of warm sodium bicarbonate for treating COVID-19. Animal studies showing efficacy of pH modulation or real-world data need to be included in the article as well as safety studies on inhaling sodium bicarbonate need to be discussed

Are sources adequately cited? Quoted or paraphrased as appropriate?
Is the review organized logically into coherent paragraphs/subsections?

No comment

Validity of the findings

Is there a well developed and supported argument that meets the goals set out in the Introduction?

No, the authors missing important aspects in the review, such as animal studies and real-world data that would point towards efficacy and safety of inhaling warm sodium bicarbonate for treating any other viral infection. Suggesting to perform “sodium inhalation in a domiciliary care setting” without enough evidence on safety and efficacy could lead to more harm than potential benefit of treating COVID-19.

Does the Conclusion identify unresolved questions / gaps / future directions?

No, based on the data available on the role of pH, temperature, osmosis and humidity for COVID-19 infection, the authors should suggest what additional studies/ gaps exist to prove that in fact modulating these cellular biochemical properties would be efficacious and safe against COVID-19 in vivo. The review doesn’t discuss any studies that need to be done and jump towards a conclusion that inhalation of warm sodium bicarbonate may help treating COVID-19 infection

Reviewer 3 ·

Basic reporting

In the review article titled ‘Extracellular pH, osmolarity, temperature and humidity can discourage SARS-CoV-2 cell docking and propagation via intercellular signaling pathways’ Cicconetti et. al. discuss various factors such as extracellular pH, osmolarity, temperature and humidity on SARS-CoV-2 cell docking and propagation, and propose warm inhalation with sodium bicarbonate as one of the therapeutic modalities in COVID19 treatment regimen.
Although very informative, the review needs serious editing and restructuring to meet PeerJ’s scientific publication standards.

Experimental design

NA

Validity of the findings

- The review is highly and unscientifically biased towards warm inhalation with sodium bicarbonate. Several key details are needed here to make this a nice review article: for example: what are other alternative approaches that are cost-effective, efficient, generally accessible in the fight against COVID19? After a brief review of these approaches, discuss how warm inhalation with SB compares to such alternative approaches.
- Authors do not provide any details regarding SB warm inhalation approach. This is a critical weakness as not all readers are aware of what SB warm inhalation methodology is.
- Authors do not provide any details regarding the current practices associated with SB warm inhalation: what is the recommended duration and frequency? What temperature? What concentration?
- Authors do not provide any details regarding whether SB warm inhalations are suggested/recommended as COVID19 preventative measure or treatment measure? Who should use it? Who should not use it?
- Authors do not discuss the limitations/disadvantages and risks associated with SB inhalation
- What are other alternatives for warm inhalations other than SB?
- Another serious scientific flaw is that with confusing sentence structures it appears that authors comparing SB inhalation to highly regulated and thoroughly tested scientific measures such as vaccination. See this sentence for example: “Thus, it might represent an additional armamentarium together with vaccination, drug repurposing, social distancing to combat this crisis that is, and remains, critical despite the enormous resources and actions already in place.”

Additional comments

Abstract: the authors provide way too much, and not directly relevant statistics related to vaccination. Please rewrite to remove all details not directly relevant to the focus of this review. Summarize ‘big picture’ scenario in no more than 1 paragraph. Include other missing, relevant and important details as highlighted previously.

Assuming that readers will refer to this review in the future; avoid phrases such as currently affecting the world. Write specific such as months and year.
“The COVID-19 pandemic caused by the SARS-CoV-2 virus is currently afflicting the world population.”

Line 58: type: ‘states’?
COVID-19 infection often leads to severe acute respiratory states.

Line 62-66: Vaccination drive is clearly not the focus of this review article. Unnecessary expansion on vaccination drive here. Delete or summarize briefly.
Stick to scientific facts and avoid subjective opinions

Line 67: As of May 15th Year is missing here, include year

Line 87-90: the wording is just off here, making the entire segment very confusing for readers.
‘may yield considerable health risk’ > does not make sense.

Line 96-97: “One of the possible approaches to modulate airway environment is the warm inhalation with sodium bicarbonate (SB)” before delving directly into warm inhalation with SB, list and discuss strengths and weaknesses associated with other approaches.

Line 138-139: Several important scientific considerations are missing here:
External pH affects several additional molecular structures such as: the fluidity of the lipid bilayer, cell shape and size, expression of membrane proteins.
Furthermore, the effects of intracellular pH are also different depending on whether the cell wall is present. Thus external pH affects mammalian cells, bacteria, viruses differently, and certainly influences microenvironment.

Line 141-142: Summarize key findings from the cited papers here (as opposed to just saying epH affect virus infection) Affects how? What type of viruses? What are the effect?

Line 160-163: No novel points are mentioned here. To make it a good review, list a few key points we need to understand to be able to develop better and more effective drugs.

Line 164-165: is there typographic errors here? The sentence doesn’t make sense.

In general: define terms at first use before abbreviating them: for example ‘IFN’
309-312:
Provide specifics. What data has been published about changes in humidity and osmolarity following SB inhalation? What durations and frequency are needed to see favorable results. If the duration and effect are not going to be physiologically relevant - there is not much strong logic in recommending SB inhalation as a measure.

Conclusion:
Who might SB inhalations be beneficial for? Is the preventative measure or a treatment option?
Dangers and limitations associated with this method?

Line 334-335: authors must not irrationally compare SB inhalations to vaccinations without significant details and analyses

How is virus progression associated with escalation of symptoms and how would SB inhalations help at different stages?

Include two figures:
1. A figure demonstrating SB inhalation
2. Add another figure outlining infections and progression of COVID19 and highlight steps where inhalation therapy helps

---

## Round 0.2 · accepted · Accept

In my view, all critiques of the reviewers were adequately addressed and the manuscript was revised accordingly. Therefore, I am pleased to accept it now.